# Role of Beneficial Microorganisms and Salicylic Acid in Improving Rainfed Agriculture and Future Food Safety

**DOI:** 10.3390/microorganisms8071018

**Published:** 2020-07-09

**Authors:** Naeem Khan, Asghari Bano, José Alfredo Curá

**Affiliations:** 1Department of Agronomy, Institute of Food and Agricultural Sciences, University of Florida, Gainesville, FL 32611, USA; 2Department of Biosciences, University of Wah, Wah Cantt 47040, Pakistan; asghari.bano@uow.edu.pk; 3Facultad de Agronomía, Departamento de Biología Aplicada y Alimentos, Universidad de Buenos Aires, Avenida San Martín 4453, Ciudad Autónoma de Buenos Aires C1417DSE, Argentina; acura@agro.uba.ar

**Keywords:** rainfed agriculture, oxidative stress, drought tolerance, phytohormones, rhizobacteria

## Abstract

Moisture stress in rainfed areas has significant adverse impacts on plant growth and yield. Plant growth promoting rhizobacteria (PGPR) plays an important role in the revegetation and rehabilitation of rainfed areas by modulating plant growth and metabolism and improving the fertility status of the rhizosphere soils. The current study explored the positive role of PGPR and salicylic acid (SA) on the health of the rhizosphere soil and plants grown under rainfed conditions. Maize seeds of two different varieties, i.e., SWL-2002 (drought tolerant) and CZP-2001 (drought sensitive), were soaked for 4 h prior to sowing in 24-h old culture of *Planomicrobium chinense* strain P1 (accession no. MF616408) and *Bacillus cereus* strain P2 (accession no. MF616406). The foliar spray of SA (150 mg/L) was applied on 28-days old seedlings. The combined treatment of the consortium of PGPR and SA not only alleviated the adverse effects of low moisture stress of soil in rainfed area but also resulted in significant accumulation of leaf chlorophyll content (40% and 24%), chlorophyll fluorescence (52% and 34%) and carotenoids (57% and 36%) in the shoot of both the varieties. The PGPR inoculation significantly reduced lipid peroxidation (33% and 23%) and decreased the proline content and antioxidant enzymes activities (32% and 38%) as compared to plants grown in rainfed soil. Significant increases (>52%) were noted in the contents of Ca, Mg, K Cu, Co, Fe and Zn in the shoots of plants and rhizosphere of maize inoculated with the PGPR consortium. The soil organic matter, total nitrogen and C/N ratio were increased (42%), concomitant with the decrease in the bulk density of the rhizosphere. The PGPR consortium, SA and their combined treatment significantly enhanced the IAA (73%) and GA (70%) contents but decreased (55%) the ABA content of shoot. The rhizosphere of plants treated with PGPR, SA and consortium showed a maximum accumulation (>50%) of IAA, GA and ABA contents, the sensitive variety had much higher ABA content than the tolerant variety. It is inferred from the results that rhizosphere soil of treated plants enriched with nutrients content, organic matter and greater concentration of growth promoting phytohormones, as well as stress hormone ABA, which has better potential for seed germination and establishment of seedlings for succeeding crops.

## 1. Introduction

Pakistan is primarily an arid country, with more than 80% area lying in arid and semi-arid regions, having an annual rainfall below 240 mm [1,2]. In Pakistan, most areas of Khyber Pakhtunkhwa receive less than 240 mm of rainfall per year. Due to this, the water becomes the most limiting factor for agriculture. In the past, resources were assigned for the improvement of irrigated areas, while no importance was given to agriculture in rainfed areas due to inherent risks in these areas [3]. However, these areas are rich in natural resources and are main contributors to the national economy. Rainfed areas endure 79% of the livestock population and are contributor to wheat (12%), maize (27%), sorghum (69%), millet (21%), mustard (25%), chickpea (77%), groundnut (90%) and to other pulse (85%) production [1,4]. These findings address an inordinate need to investigate the adverse effects of moisture deficit in rainfed areas on plant growth and its ameliorative measures to enhance plant yield under these conditions. Most of these approaches are costly and time consuming. Recent studies demonstrate that the microbes alone or in combination with salicylic acid can significantly improve the overall growth and productivity of crop plants grown in rainfed areas [5,6].

Bacteria that colonize plant roots and enhance plant growth are stated as plant growth-promoting rhizobacteria (PGPR). PGPR are an extremely diverse group of micro-organism that occur in the vicinity of plant roots [7]. They significantly influence plant health as they enhance plant tolerance to various stresses and improve the fertility status of rhizosphere soil. Rhizosphere soil acts as a fount of all nutrients needed by plants for growth. The three most common nutrients are nitrogen, phosphorus and potassium, followed by calcium, magnesium and sulfur. Plants also need small quantities of iron, manganese, zinc, copper, boron and molybdenum [8]. However, stressful environments result in unusually low or high soil pH, which in turn result in the unavailability of certain nutrients to the plants. PGPR improves this availability and enhances the mobilization of macro and micronutrients from the rhizosphere soil to the host plant. They are well-known to mend the effectiveness of plants and respond to various stimuli [9]. PGPR-induced proliferation of root systems and improvement in root architecture have been the primary effects [10,11].

Previous studies have reported that PGPR affects the physiology of plants to attenuate the adverse effects of both the biotic and abiotic stresses [12,13]. These microorganisms improve water use efficiency and regulate photosynthesis through modulating the stomatal opening and hydraulic conductivity of the roots. Marulanda et al. [14] reported that maize plants inoculated with *Bacillus* sp. showed an increase in root hydraulic conductivity, as compared to uninoculated plants grown under stress conditions. Colonization by PGPR is associated with changes in plant metabolism, signaling and hormone homeostasis. Different PGPR strains can synthesize phytohormones, metabolize them, or affect plants’ hormone synthesis and signal transduction [15]. PGPR that produce auxins have been shown to elicit transcriptional changes in hormone, defense-related and cell wall-related genes [16], induce longer roots [17], increase root biomass and decrease stomata size and density [18] and activate auxin response genes that enhance plant growth [19]. Some strains of PGPR can promote relatively large amounts of gibberellins, leading to enhanced plant shoot growth [20]. Interactions of these hormones with auxins can alter root architecture [21]. PGPR induce changes in phytohormone signaling and osmolyte accumulation facilitates plants to grow well under stressful environments. Inoculation with *B. subtilis* increases photosynthesis in *Arabidopsis* through the modulation of plant endogenous sugar and abscisic acid (ABA) signaling, with a regulatory role for plant symbionts in photosynthesis [22]. Shi et al. [23] showed that endophytic bacteria species increased the photosynthetic capacity and total chlorophyll content of sugar beet, leading to a consequent increase in carbohydrate synthesis; these increases were promoted by phytohormones, which were produced by the bacteria. Besides this, beneficial microorganisms can stimulate carbohydrate metabolism and transport, which directly implicate source-sink relations, photosynthesis, growth rate and biomass reallocation. Seed inoculated with *B. aquimaris* strains increased total soluble sugars and reduced sugars in wheat under saline field conditions and resulted in higher shoot biomass and NPK accumulation [24]. PGPR also play an important role in enhancing the fertility status of rhizosphere through various mechanisms, such as decomposition of crop residues, mineralization of soil organic matter, immobilization of mineral nutrients, phosphate solubilization, synthesis of soil organic matter, nitrification, nitrogen fixation, plant growth promoters including nutrient acquisition phytohormone production (biostimulants) and suppression of plant pathogens [25]. In contrast to PGPR, salicylic acid is a chemical messenger, significantly affecting the plant responses and water budget under stressful environments.

Maize (*Zea mays* L.) belongs to the grass family Gramineae and an important cereal crop of the globe. In Pakistan, maize is gaining a significant position because of its high yield potential, greater biomass for cattle and short life cycle [26]. It is an essential component of global food security, cultivated with an area of above 118 million hectares with yearly production of 600 million metric tons. Maize, along with wheat and rice, gives about 30% of the food calories to 4.5 billion people in developing countries and 67% of the total maize is grown in these countries; hence, maize plays an important role in the livelihoods of millions of poor people [27]. The present study was based on the hypothesis that PGPR isolated from moisture stressed habitat can better impart tolerance to plant against stresses when used as bioinoculant. Combined treatment with a plant growth regulator (SA) may be more beneficial to improve the defense system of plants against diseases. SA induces systemic acquired resistance, and is hence expected to enhance ISR potential of PGPR. It is speculated that the residual effects of PGPR and SA may also persist in the rhizosphere soil to improve the seedling establishment of the succeeding crop. Hence, the present investigation was made to evaluate the effects of PGPR isolates (from drought affected areas) and SA alone and in combination on the physiology and growth of two varieties of maize differing in drought tolerance.

## 2. Material and Methods

### 2.1. Experimental Work Plan

The maize plants were grown under natural rainfed conditions in a field at Kohi Barmol, located 35 km from Mardan. Mardan is located in Pakistan, in the south western part of Khyber Pakhtunkhwa Province (at 34°12′0 N 72°1′60 E and at an altitude of 283 m). Seeds of two maize varieties, i.e., SWL-2002 (drought tolerant) and CZP13-2001 (drought sensitive) were obtained from Ayub Agriculture Research Institute, Faisalabad. The inoculated maize seeds were sprayed with an aqueous solution of SA (150 mg/L) at three leaf stage. Salicylic acid is a beta hydroxy acid that occurs as a natural compound in plants and is hormonal in action. The experiment comprised five treatments (Table 1) and was conducted using randomized complete block design (RCBD) with a plot size of 5 × 1.5 m, and a row-to-row distance of 40 cm. Four replicates were used per treatment.

### 2.2. Seed Sterilization and Inoculation

Maize seeds were surface sterilized with ethanol (70%) for 3 min, followed by shaking with n Clorox (10%) for 3 min. The seeds were successively washed in autoclaved distilled water followed by soaking in broth culture of PGPR for 3 h prior to sowing. There were 1 × 10^5^ colonies of bacteria per seed.

### 2.3. Physiological Characterization of Plant

For all physiological activities, the plant tissues were collected after one week of SA application.

#### 2.3.1. Leaf Chlorophyll and Carotenoids Content

Chlorophyll content was calculated with the help of the soil plant analysis development (SPAD) chlorophyll meter. In each plant, the chlorophyll content of four leaves was averaged and used as single replication. The carotenoids content was determined following the method of Lichtenthaler and Welburn [28].
$$Carotenoids\;\left(\frac{mg}{g}\right)=Absorbance\;\left(OD_{663}\right)\times 4$$

#### 2.3.2. Chlorophyll Fluorescence and Performance Index

The chlorophyll fluorescence and performance index were used to assess the photosystem II efficiency. The chlorophyll fluorescence was measured using pulse modular fluorometer. The trait was measured on intact leaves of the abaxial surface (third leaf) after 30 min of dark adaptation [29]. Plant vitality was characterized by performance index parameter [30] calculated as follows:$$PI=1-\frac{\left(F0-FM\right)}{M0-VJ}\;\times \;\frac{FM-F0}{F0}\;\times \;\frac{1-VJ}{VJ}$$
where: *F*_0_ is fluorescence intensity at 50 µs, *F_J_* is fluorescence intensity at the J step (at 2 ms), *F_M_* represents maximal fluorescence intensity and *V_J_* is relative variable fluorescence at 2 ms calculated as *V_J_* = (*F_J_* − *F*_0_)/(*F_M_* − *F*_0_), *M*_0_ represents initial slope of fluorescence kinetics, which can be derived from the equation: *M*_0_ = 4 ✕ (*F*_300_ µs − *F*_0_)/(*F_M_* − *F*_0_).

#### 2.3.3. Leaf Proline Content and Lipid Peroxidation

Leaf proline content was determined following the method of Bates et al. [31], whereas Li’s [32] method was followed for determination of lipid peroxidation. The lipid peroxidation was determined by calculating the amount of Malondialdehyde (MDA) formed by thiobarbituric acid (TBA) reaction. The MDA concentration was determined by the following formula:

C_MDA_ (μ mol L^−1^) = 6.45 (A_532_ − A_600_) − 0.56 A_450_, from which the absolute concentration (mmol g¡1 FW) of malondialdehyde was calculated.

#### 2.3.4. Antioxidant Enzymes

We also determined the activities of peroxidases [33], catalases [34] and superoxide dismutases [35] in shoots of maize plants.

#### 2.3.5. Determination of IAA, GA and ABA Contents of Maize Leaves and Rhizosphere Soil

Plant hormones were extracted following the method of Kettner and Dörffling [36]. Fresh plant leaves (1 g) were grinded in 80% methanol at 4 °C with butylated hydroxytoluene @ 10 mg/L used as an antioxidant. The samples were analyzed on high performance liquid chromatography (HPLC). For the identification of hormones, 100 µL samples filtered through Millipore (SPEC) filter were injected into the column. Commercial-grade abscisic acid (ABA), indole acetic acid (IAA) and gibberellic acid (GA), were used as reference for the retention time and peak area measured at 280 nm and 254 nm, respectively. For ABA, the samples were injected on to a C_18_ column and eluted at 254 nm with a linear gradient of methanol (30–70%), containing 0.01% acetic acid [37], whereas the phytohormones contents of soil were analyzed using the method of Hartung et al. [38]. For this purpose, the soil samples were extracted in threefold excess of 1 Mm CaCl_2_ for an hour. The extraction was then partitioned thrice with ethyl acetate followed by drying of ethyl acetate with the help of rotary thin film evaporator (RFE). The dried sample was dissolved in 1mL methanol (100%) and analyzed for the presence of phytohormones using HPLC as described above.

#### 2.3.6. Relative Water Content and Soil Moisture Content

Relative water content (RWC) of leaves was determined following the formula suggested by Barrs and Weatherly [39]. Leaf RWC was determined with fully expanded leaves by clipping and weighing the fresh leaves. The leaves were then placed in a Ziploc bag filled with distilled water for 24 h at room temperature and then the turgid weight was immediately calculated after excess moisture was removed. The leaves were then dried in an oven at 60 °C for 72 h to determine the leaf dry weight. Percent moisture content was calculated as 100 × [(*W*1 − *W*2)/*W*2], where *W*1 is fresh soil weight and *W*2 is oven dried weight of 3 kg of field soil.

### 2.4. Soil Analysis

For soil analysis, soil samples were collected from the rhizosphere of maize at a depth of 10 cm and packed in a Ziploc bag. The collected samples were sent to the laboratory within a few hours and stored at −80 °C. The collected soil was analyzed for different micro (Cu, Co, Fe and Zn) macronutrients (Ca, Mg, K and Na) following the ammonium bicarbonate-DTPA method developed by Soltanpour and Schwab [40]. The method of Nelson and Sommers [41] was employed for carbon and total soil nitrogen determination.

### 2.5. Nutrient Analyses of Plant Extracts

Oven dried leaf samples (0.25 g) were taken in 50 mL flask to which 6.5 mL of mixed acid solution, i.e., nitric acid, sulphuric acid, perchloric acid (5:1:0.1), were added and boiled on hot plate under fume hood until the digestion was completed, which was indicated by white fumes coming out from the flasks. Thereafter, a few drops of distilled water were added and allowed to cool. The digested samples were transferred in 50 mL volumetric flasks and the volume was made up to 50 mL by adding distilled water. The extract was filtered with Whatman No. 42 filter paper and concentration of these elements was determined by the atomic absorption spectrophotometer (Shimadzu AA-670).

Stock solution (100 ppm) of different elements were prepared for the determination of macro and micronutrients following the method of Allen et al. [42].

### 2.6. Data Analysis

The data were analyzed using the statistical software package IBM SPSS Statistics V23. Before carrying out statistical tests, normality of the data was checked by means of the Kolmogorov–Smirnoff statistic (*p* > 0.01).

## 3. Results

### 3.1. Leaf Chlorophyll and Carotenoids Contents

In general, the chlorophyll content was decreased significantly in both the sensitive (V1) and tolerant varieties (V2) under rainfed conditions, as compared to the irrigated control (Figure 1). The chlorophyll content of the drought sensitive variety was 43% lower than the control under rainfed conditions. The percent decrease was completely overcome by the application of PGPR and SA, alone or in combination. The combined treatment of PGPR and SA (consortium) significantly improved the chlorophyll content of sensitive variety and reduced the damage to 3%. Similarly, only PGPR treatment was also effective in both the sensitive and tolerant varieties and significantly reduced (<9%) the percent damage to leaf chlorophyll content. SA alone was also effective in reducing the percent damage and enhanced (32% and 12%, respectively) the content of chlorophyll in both the varieties. The tolerant variety was more resistant to moisture stress under rainfed conditions and showed 26% higher chlorophyll content than the sensitive variety. A similar pattern of response was also recorded for the carotenoids content in the shoot of both the varieties treated with PGPR and SA, alone or in combination.

### 3.2. Chlorophyll Fluorescence and Performance Index

In general, the chlorophyll fluorescence decreased significantly in both the sensitive (V1) and tolerant variety (V2) under rainfed conditions, as compared with the irrigated control (Table 2). However, the percent damage was less in the tolerant variety than the sensitive variety under all treatments. The damage to chlorophyll fluorescence of the sensitive variety was more (32%) than the tolerant variety, however; the application of PGPR and SA significantly enhanced (52%) the chlorophyll fluorescence of the sensitive variety even under rainfed conditions followed by the application of PGPR alone (42%). The combined treatment of PGPR and SA further augmented the percent inhibition. The increases in chlorophyll fluorescence in both the varieties due to combined treatment of PGPR and SA was even greater than the irrigated control. SA alone was also effective and significantly enhanced (40% and 20%, respectively) the chlorophyll fluorescence of both the varieties in comparison to uninoculated untreated plants grown under rainfed conditions.

The performance index (PI) showed significant differences within treatments and varieties grown under rainfed conditions. Rainfed conditions decreased the PI of both the varieties; however, the percent damage was more (35%) in sensitive variety than that of the tolerant variety. The consortium of PGPR and SA was most effective in reducing the percent damage and significantly improved the PI of both the sensitive (49%) and tolerant variety (25%), as compared to untreated uninoculated plants grown under rainfed conditions. Salicylic acid was also effective and improved the PI of both the varieties (36% and 16%, respectively) grown under rainfed conditions (Table 2).

### 3.3. Antioxidant Enzymes Activities

The uninoculated untreated plants grown under rainfed conditions showed higher values for antioxidant enzymes activity in comparison to irrigated control (Figure 2). PGPR inoculation and application of SA significantly lowered the activities of antioxidant enzymes in both the varieties, in comparison to plants grown under rainfed conditions. The consortium of PGPR and SA was the most efficient treatment and significantly reduced the activities of catalase (>30%), POD (>36%) and APOX (>60%) in both the sensitive and tolerant varieties; however, the treatment was more responsive in the sensitive variety for reducing the activities of catalase and APOX. The PGPR treatment significantly reduced (61% and 58%, respectively) the activities of APOX in both the sensitive and tolerant varieties.

### 3.4. Leaf Proline Content and Lipid Peroxidation in the Shoot of Plants

Rainfed conditions significantly increased leaf proline content. The tolerant variety (V2) showed a higher increase (10%) in proline content than the sensitive variety (V1). Treatment of plants with the consortium of PGPR and SA significantly lowered (27% and 59%, respectively) the proline production and brought the values very close to the irrigated control in both the sensitive and tolerant varieties. The foliar application of SA was more effective than the combined treatment of 2-PGPR, i.e., *P. chinense* and *B. cereus*, and significantly reduced the proline production in the sensitive variety. Similarly, rainfed conditions alleviated the level of lipid peroxidation in both varieties, but the increase was more pronounced (37%) in tolerant variety than sensitive variety. Maximum decrease (27% and 51%) in lipid peroxidation was noted in plants of both the varieties receiving the foliar application of SA. Application of consortium of PGPR and SA on plants grown under rainfed conditions also reduced (32% and 25%) the alleviated level of lipid peroxidation followed by the PGPR treatment (i.e., *P. chinense* and *B. cereus*), but both of these treatments were more responsive in the sensitive variety compared to the tolerant variety (Figure 3).

### 3.5. IAA Content in Plant Shoot and Rhizosphere Soil

Rainfed conditions significantly reduced the IAA concentration in both the shoot and rhizosphere soil of drought sensitive and tolerant varieties; however, the percent damage was more significant in the sensitive variety than the tolerant variety (Figure 4). The tolerant variety was able to maintain higher IAA content (46% and 89%) in both the shoot and rhizosphere soil. The consortium of PGPR and SA was stimulatory to IAA and significantly enhanced the content of IAA of both the varieties in shoot (73% and 56%) and in rhizosphere soil (93% and 45%), respectively. The effect of the consortium of PGPR and SA was on par with PGPR treatment in sensitive variety. Foliar application of SA was also effective and enhanced the IAA content in both the shoot and rhizosphere; even the increase due to SA was greater than the PGPR treatment (i.e., *P. chinense* and *B. cereus*).

### 3.6. ABA Content in Plant Shoot and Rhizosphere Soil

Rainfed conditions resulted in a significant increase in the content of ABA of the drought sensitive variety; however, the application of PGPR alone or in combination with SA significantly reduced the ABA content (Figure 5). The most significant decrease (>55%) in the content of ABA under rainfed conditions was noted in the shoot of plants treated with the consortium of PGPR and SA followed by PGPR alone. Salicylic acid applied singly significantly reduced the ABA content in the shoot of sensitive variety. Similarly, the pattern of response was noted in the rhizosphere soil of both the varieties grown under rainfed conditions.

### 3.7. GA Content in Plant Shoot and Rhizosphere Soil

The GA content in the shoot and rhizosphere soil of both the varieties was significantly reduced in untreated plants grown under rainfed conditions (Figure 6). However, the application of PGPR and SA was more effective and significantly enhanced (>70%) the content of GA in the shoot and rhizosphere soil of both the varieties. Inoculation of plants with PGPR without SA also resulted in significant increase in the concentration of GA and it had an equal effect in the rhizosphere of both the varieties. The consortium of PGPR and SA was more effective in the sensitive variety whereas, PGPR treatment was more effective in the tolerant variety. The foliar application of SA alone was more stimulatory in the sensitive variety and enhanced the GA content by 81% in both the shoot and rhizosphere, whereas the increase in the tolerant variety was 67%, compared to untreated uninoculated plants grown under rainfed conditions.

### 3.8. Micronutrient Accumulation in Shoot and Rhizosphere Soil

The accumulation of all studied micronutrients was adversely affected in both the shoot and rhizosphere soil of plants grown under rainfed conditions, but the percent decrease was higher in the sensitive variety (Table 3 and Table 4). The PGPR, SA and combined treatment of PGPR and SA not only alleviated the inhibitory effects of low moisture stress, but also augmented the accumulation of the metals. The combined treatment of PGPR and SA enhanced the accumulation of Cu (1.9-fold and 2.3-fold), Co (1-fold and 4.7-fold), Fe (0.8-fold and 3.8-fold) and Zn (80% and 54%), respectively, in the shoot of the drought sensitive and the tolerant variety. The same was found most effective for the accumulation of micronutrients in the rhizosphere soil of both the varieties. The sensitive variety responded more. Salicylic acid alone assisted plants in the accumulation of micronutrients. This treatment (SA) was much responsive for Cu accumulation (97%) in the tolerant variety and for Fe accumulation (91%) in the sensitive variety.

### 3.9. Macronutrient Accumulation in Shoot and Rhizosphere

All the treatments significantly enhanced the macronutrient accumulation in the shoot and rhizosphere of maize grown under rainfed conditions (Table 5 and Table 6). The combined treatment of 2—PGPR (*P. chinense* and *B. cereus*) was more effective and resulted maximum increase in Ca (52% and 88%) and K (63% and 99%) accumulation in both the shoot and rhizosphere soil of the sensitive variety and was more effective for the increase in Mg content (69% and 73%) of the tolerant variety. The combined treatment of PGPR and SA (consortium) was more effective for Ca (71% and 99%) and Na (80% and 93%) accumulation in the shoot and rhizosphere soil of the sensitive variety and also for Ca (71% and 99%), K (54% and 98%) and Na (69% and 71%) content in the shoot and rhizosphere of the tolerant variety. The % increase in the Ca content of the rhizosphere of both the sensitive and tolerant varieties was similar in plants treated with the consortium of PGPR and SA. Application of SA further augmented the content of macronutrients in both plant shoot and rhizosphere. Maximum increase (68% and 48%) due to the application of SA was noted in the Ca and Na contents in the shoot of both the sensitive and tolerant variety. Salicylic acid application was more effective in the sensitive variety.

### 3.10. Relative Water Content (RWC) and Soil Moisture Content (SMC)

Drought stress resulted in significant decrease in RWC and SMC of the sensitive variety, whereas the decrease in the tolerant variety was lower (Table 7). The RWC and SMC of the sensitive variety (V1) was decreased by 55% and 86% under the rainfed conditions compared with the control condition, respectively, but the application of PGPR and SA significantly reduced the percent decrease caused by rainfed condition and improved the RWC (52%) and soil moisture content (86%) of the sensitive variety. The treatment of PGPR in combination with SA was more effective for RWC in the sensitive variety. The soil moisture content of the tolerant variety was higher in all the treatments than sensitive variety. The application of SA alone improved the RWC (29%) and SMC (80%) of the sensitive variety, whereas it had no significant effect in the tolerant variety.

### 3.11. Soil EC and pH

The electrical conductivity (EC) was higher in the rhizosphere soil of the tolerant variety than that of the sensitive variety, except in PGPR treatment (Table 7). Under rainfed conditions, the soil pH was decreased in both the varieties. The higher values of EC in the sensitive variety was noted in the rhizosphere of plants treated with the combination of 2-PGPR, followed by the combined treatment of PGPR and SA. However, in the tolerant, a variety of the higher values of electrical conductivity was noted in the rhizosphere soil of plants treated with PGPR and SA. Overall, most of the treatments showed no significant differences in the EC of soil samples. The pH of the sensitive variety was more affected, as compared to tolerant variety. Maximum soil pH (8.7 and 9.2) was noted in the rhizosphere soil of sensitive and tolerant varieties treated with PGPR, followed by the combined treatment of PGPR and SA (Table 7).

### 3.12. Soil Organic Carbon Content (SOC) and Total Nitrogen in the Rhizosphere of Sensitive and Tolerant Variety

The present study highlighted that higher concentration of SOC and nitrogen content occurred in the irrigated control (Table 8). The concentration of SOC was increased by 43% and 42% in the rhizosphere of both the sensitive and tolerant variety when treated with the consortium of PGPR and SA followed by PGPR treatment made alone. In most of the treatments, the SOC content was found in medium range (1.5–3.9). Similarly, a higher concentration (43% and 36%) of TN was noted in the rhizosphere soil of both the varieties treated with the consortium of PGPR and SA (Table 8).

### 3.13. C/N Ratio and Soil Bulk Density

The carbon-to-nitrogen ratio (C/N) is an important factor affecting the overall turnover rates of soil organic matter. The higher (42% and 26%) carbon-to-nitrogen ratio was found in plants receiving the combined treatment of PGPR and SA in both the varieties. The combined treatment of 2-PGPR was also effective and significantly enhanced the C/N ratio in the rhizosphere of both the varieties. In contrast to the C/N ratio, the soil bulk density was increased under rainfed conditions. However, the application of PGPR and SA significantly reduced the soil bulk density and enhanced the soil organic carbon and C/N ratio. Maximum decrease (42%) in soil bulk density was noted in the rhizosphere of plants treated with the consortium of PGPR and SA of both the varieties. The combined treatment of 2PGPR was also effective in reducing the soil bulk density and increasing the soil organic matter. Salicylic acid alone was also effective in enhancing the C/N ration and decreased the soil bulk density in the rhizosphere of both the varieties (Table 8).

## 4. Discussion

Rainfed agriculture comprises more than 75% of the cultivated area of the world. In Pakistan more than 25% of the total area is designated as rainfed and more than 30% population is dependent on rainfed agriculture. Furthermore, these regions are badly affected by frequent droughts that adversely affect plant physiology and productivity, consequently affecting the livelihood of the inhabitant. Low productivity in these regions is accompanied by limited water supply, degraded soil health associated with low fertility and limited supply of nutrients. The present investigation yielded valuable information pertaining to the phytohormonal status of soil which originates from the root exudates of the growing plants treated with PGPR and SA. This information may be useful for the establishment and proliferation of successive crops exposed to moisture deficit conditions.

Rainfed conditions adversely affected the growth of plants and photosynthetic efficiency, as evidenced by the significant decrease in the chlorophyll and carotenoids contents, chlorophyll fluorescence and performance index (PI). The sensitive variety was affected more. Previous studies document significant reductions in these parameters in plants grown under rainfed conditions [43]. The ameliorative effects of PGPR were more pronounced than SA for the increase in the contents of chlorophyll and carotenoids. The observed stimulatory effects were further augmented in the combined treatment and may be attributed to the fact that PGPR and SA significantly improved the relative water content of the plants grown under rainfed condition, demonstrating the positive relationship between RWC and chlorophyll fluorescence. The effect was more pronounced in the sensitive variety and the plants were more responsive to the combined treatment of consortium of PGPR and SA. Notably, the drought-induced decrease in carotenoids content was much greater in the sensitive variety than tolerant variety. Carotenoids act as protective pigment and scavengers of ROS [44], thereby enhancing the chlorophyll fluorescence and performance index. As potential antioxidants, they are essential in different plant processes under stress. They act as light harvesters and scavenge the triplet state chlorophylls and singlet oxygen species, depleting excess harmful energy during stress conditions and membrane stabilizers [45]. Carotenoids also play an important role in the mechanisms protecting the photosynthetic apparatus against various harmful environmental factors [46], thus ensuring enhanced plant growth under unusual circumstances. SA was considered to prevent the degradation of photochemical pigments under stress conditions [47,48]. The higher chlorophyll fluorescence is an indicator for detection of plant tolerance to various stresses. PI is a sensitive indicator of water scarcity in plants [49,50]. The applications of SA and PGPR assisted plants in water and nutrient uptake and thus maintained higher chlorophyll fluorescence and PI under rainfed conditions. The combined treatment of PGPR and SA significantly improved the water budget of plants, which eventually ensured higher photochemical efficiency and growth of plants of both the varieties, compared to untreated plants [51,52,53,54].

An increased level of antioxidant enzymes, such as CAT, POD and APOX, was exhibited in the current study, demonstrating better potential for scavenging the reactive oxygen species (ROS) also reported previously in wheat [55], barley [56], rice [57], soybean [58] and chickpea [59] under drought conditions. The PGPR and SA significantly reduced the activities of antioxidant enzymes, as compared to untreated plants grown under rainfed conditions. This decrease could be attributed to the fact that application of PGPR in combination with SA overcame the oxidative stress, which is produced as secondary stress under drought stress; thereby, the production of ROS is minimized and, hence, the production of extra antioxidant enzyme activity may not be required. Khan and Bano [60] demonstrated a PGPR-induced reduction in the activities of antioxidant enzymes in chickpea shoots grown in sandy soil conditions. Reduction in antioxidant enzymes activity by the application of SA had also been reported previously [61]. PGPR and SA decreased the lipid peroxidation, as measured by the malondialdehyde content. The content of lipid peroxidation increases in plants with the increase in ROS, thereby adversely affecting the physiological and biochemical process in plants under stress. Higher content of lipid peroxidation also adversely affects permeability of cell membranes, ion transport, enzymatic activity and protein cross-talk, thus disrupting overall cellular metabolism and eventually leading to cell death [62]. PGPR reducing oxidative damages by reducing lipid peroxidation had been reported earlier by Habibzadeh et al. [63] in canola and by Sahin et al. [64] in lettuce plants grown under waterlogged conditions.

Plant growth and development is controlled by phytohormones and the PGPR-based direct promontory effects involved in the biosynthesis of IAA and GA in plants [65]. Drought-induced decrease in IAA was higher in the sensitive variety as compared to control. The tolerant variety resisted the adverse effect of drought to a greater extent. All the treatments have ameliorative effects but SA treatment showed lower potency in this context, which could be due to the basic difference between the PGPR and SA in the biosynthesis or modulation of phytohormones. PGPR-induced increase in IAA was greater in the sensitive variety. A similar pattern of response was recorded in the GA content of rhizosphere soil. The rhizosphere soil demonstrated the residual effects of growing drought stressed plants and PGPR treated plants. The drought-induced decrease in IAA content was alleviated by PGPR. The PGPR consortium assisted SA in combination. GA is an important plant growth promoting hormone responsible for cell elongation and subsequently higher biomass production [66]. GA content was several times higher in plants treated with PGPR alone or more so in consortium. This is an adaptive mechanism to combat the adverse effect of drought (moisture stress) and promote the growth and biomass production in plants. The SA was less effective than the PGPR/PGR consortium. Phytohormones produced by root-associated microbes may be an important target for metabolic engineering host plants to induce tolerance to abiotic stresses.

As documented previously, in all stresses, the endogenous ABA level increases significantly to impart tolerance to stresses [67]. Notably, the drought-induced increase in ABA content was higher in the sensitive variety. The PGPR and SA effects of decreasing drought-induced augmentation in ABA were higher in the tolerant variety; this may be attributed to the fact that drought-induced decrease in soil moisture and RWC was less in the tolerant variety. The significant increase in ABA content in plants under rainfed conditions, particularly in the sensitive variety, was counteracted by PGPR and SA application. The PGPR used were EPS producing bacteria and also function in water conservation and, hence, reduced significant increase in ABA as higher ABA exerts inhibitory effects on plant productivity and stomatal conductance was reduced [68,69]. However, some increase in ABA is required to combat stress, as ABA also acts as a stress hormone and ameliorates the osmotic and oxidative stresses in plants which are created under moisture deficit conditions in soil. ABA is considered to accumulate in the roots and transported along with the xylem sap to the leaves [70]. However, the PGPR, particularly the consortium of PGPR in combination with SA, enhance the ABA content more than the control; this aimed to improve the water holding capacity of soil and to minimize the evapotranspiration and the water loss [67]. The effect of PGPR inoculation and that of SA was at par, but the consortium was even more stimulatory and the percent increase was higher in the tolerant variety and correlated with the soil moisture retained in the tolerant variety. The phytohormone analysis of rhizosphere soil demonstrated the significant increase in ABA content, possibly originating from the root exudate of stressed plants, in addition to being synthesized by the associated microbes. This warrants the inhibition in germination and seedling establishment of the succeeding crop. Nevertheless, the rhizosphere of PGPR or SA treated plants exhibited a decrease in drought-induced ABA accumulation. Thus, the soil can have better potential for seed germination and seedling establishment as consortium increased IAA, GA content, while it decreased the ABA content.

Relative water content of leaves (RWC) and soil moisture content (SMC) are indicators of the water status of plant and rhizosphere soil, respectively, under drought stress conditions [71]. The application of PGPR alone and more so in combination with SA alleviated the decrease in RWC and SMC and significantly improved the RWC and SMC of both the varieties grown under rainfed conditions. Higher RWC enables plants to recover from stresses and improve its growth and yield stability [72,73]. Kasim et al. [74] and Mahmood et al. [75] reported significant increases in the RWC of barley plants inoculated with biofilm forming plant growth promoting rhizobacteria. Availability of soil moisture is a prerequisite for seed germination, root development and nutrient uptake [76]. Naveed et al. [77] reported that bacterial inoculation increases the relative water content in plants and decreases the electrolyte leakage leading to drought tolerance in plants. The moisture deficit condition of rainfed soil decreases soil moisture which was alleviated and taken as par to the control by the application of a consortium of PGPR and SA, which was more pronounced in the tolerant variety (V2).

PGPR releases various chelating substances in the rhizosphere that affect the availability and mobility of essential nutrients [78]. The inoculation of plants with. *P. chinense* and *B. cereus* significantly improved the accumulation of macro and micronutrients in plant shoot and rhizosphere soil of both the varieties. The PGPR was more effective and assisted SA when in combination and was more effective and the pronounced increase was recorded in the contents of Ca, K, Mg, Na, Cu, Co, Fe and Zn in the rhizosphere soil and subsequently in shoots of plants under rainfed conditions, the tolerant variety being more responsive. This is further evidenced by the result that the basic level of all the macro and micronutrients was higher in the tolerant variety over that of the sensitive variety, though the sensitive variety was more responsive to treatments. Iron and zinc are integral parts of some enzymes and pigments and play an important role in the synthesis of DNA, photosynthesis and respiration, and assist in the production of energy in plants under extreme climatic conditions [79]. Higher content of Cu in the rhizosphere involved in lignification provides strength and prevents lodging in cereal crops; the deficiency of Cu in plants leads to poor growth, delayed flowering and sterility, and also enhances the susceptibility to diseases [80]. The combined treatment of 2-PGPR augmented Ca, K and Mg accumulation in the shoot, as well as in the rhizosphere of the sensitive variety. Rana et al. [81] reported a significant increase in the nutrient content of plants inoculated with the consortia of 3-PGPR. Aini et al. [82] reported that the inoculation of lettuce plants with PGPR and AMP resulted in improved root colonization and macronutrient uptake.

Soil organic matter is the index of soil quality and soil stability [83]. Rainfed conditions adversely affected the soil organic content and nitrogen in the rhizosphere of untreated plants. PGPR and SA significantly improved the soil organic content and total nitrogen in the rhizosphere of both the varieties, but the tolerant variety was more responsive. Decomposition of organic matter results in the release of essential nutrients and thus helps in maintaining nutrient cycling. A higher concentration of soil organic content assists in binding soil particles into aggregates, which facilitates water filtration and aeration. Soil organic matter are the major pools of carbon in the biosphere and can act both as a source and sink of carbon [84,85]. Previously, Hassan and Bano [86] reported that the inoculation of wheat plants with PGPR significantly improved the soil organic content and total nitrogen in rhizosphere soil. Present results are in agreement with those reported by Hassan and Bano [86] in wheat, Xiaohui et al. [87] in tomato plants and Naseri et al. [88] in rapeseed, where the application of PGPR significantly enhanced the soil nitrogen and organic matter. This significant increase in the consortium of PGPR and SA was due to their ability to marinate higher C/N ratio in soil as C/N ratio affects overall turnover rates of soil organic matter and improves soil structure and plant growth.

## 5. Conclusions

The PGPR and, more effectively, combined treatment of the consortium of PGPR and SA improved the growth of maize, but also improved Ca, K, Mg, Zn and Fe in shoot and rhizosphere soil. The PGPR alone and more so in consortium with SA not only induced tolerance in plants to moisture stress, but also exerted positive residual effects in the rhizosphere soil. Conservation in soil moisture content, reduced evapotranspiration, increased organic matter, high C/N ratio and decrease in bulk density are prerequisites for growth proliferation of the succeeding crops. PGPR-induced modulation of growth promoting hormones, particularly IAA and GA, and decreased in ABA content of plants and rhizosphere soil, providing another alma mater for improved growth and higher yield in addition to enhanced soil fertility status. SA alone is not very effective, but its combined treatment is postulated with PGPR consortium preferably having EPS production potential.

## Figures and Tables

**Figure 1 microorganisms-08-01018-f001:**
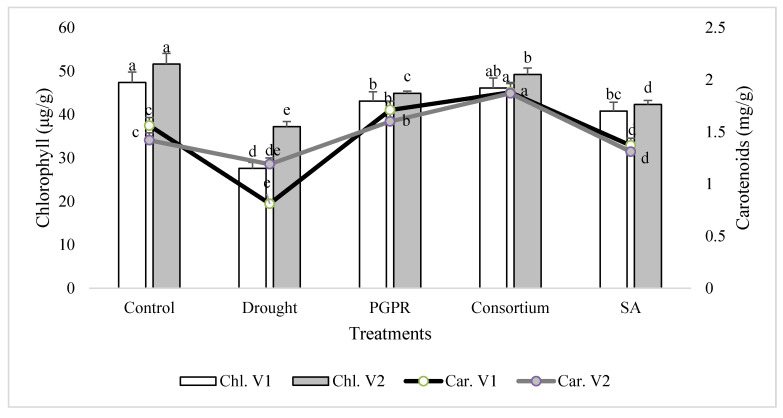
Leaf chlorophyll and carotenoids contents of drought sensitive (V1) and tolerant varieties (V2) grown under rainfed conditions. PGPR: *Planomicrobium chinense* and *Bacillus cereus*; SA: salicylic acid. Chl: chlorophyll; Car: carotenoids. Different significance levels were denoted with different letters (a, b, c, d and e).

**Figure 2 microorganisms-08-01018-f002:**
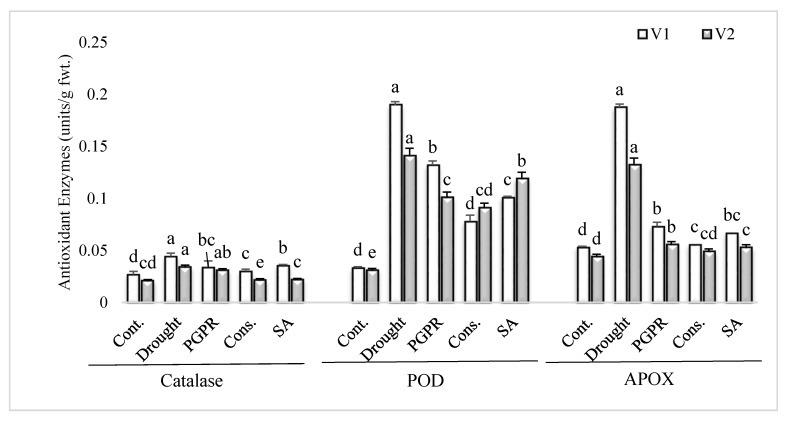
Antioxidant enzymes activity in the leaves of drought sensitive and tolerant varieties grown under rainfed conditions. V1: drought sensitive variety; V2 drought tolerant variety. Different significance levels were denoted with different letters (a, b, c, d and e).

**Figure 3 microorganisms-08-01018-f003:**
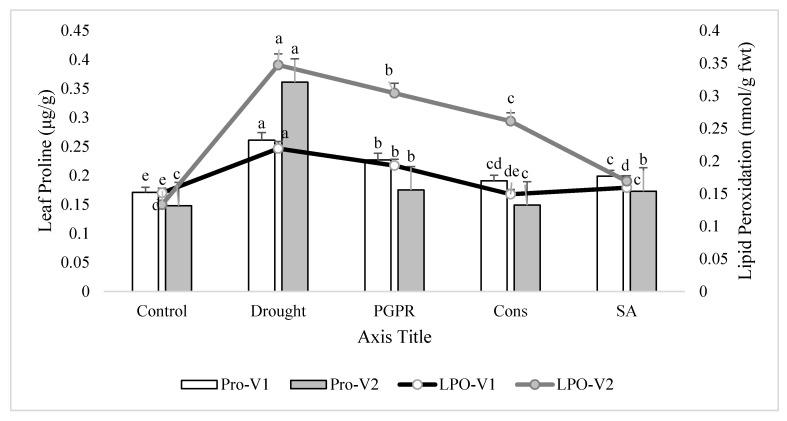
Leaf proline content and lipid peroxidation in the leaves of drought sensitive and tolerant varieties grown under rainfed conditions. V1: drought sensitive variety; V2: drought tolerant variety; Pro: proline; LPO: lipid peroxidation. Different significance levels were denoted with different letters (a, b, c, d and e).

**Figure 4 microorganisms-08-01018-f004:**
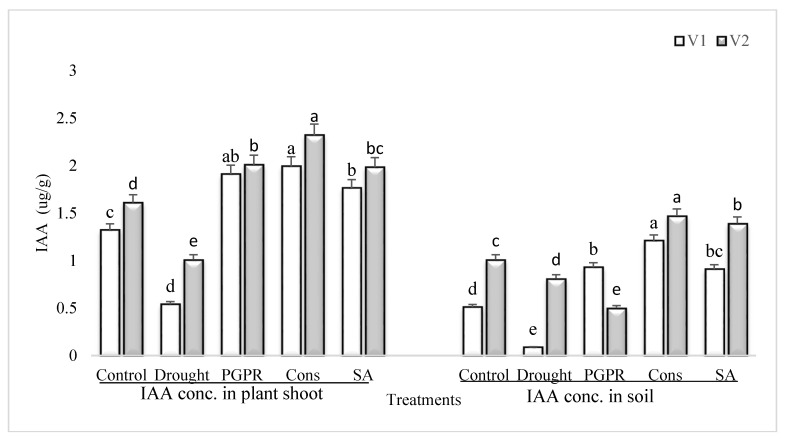
IAA content in plant shoot and rhizosphere of drought sensitive and tolerant varieties grown under rainfed conditions. V1: drought sensitive variety; V2: drought Tolerant variety. Different significance levels were denoted with different letters (a, b, c, d and e).

**Figure 5 microorganisms-08-01018-f005:**
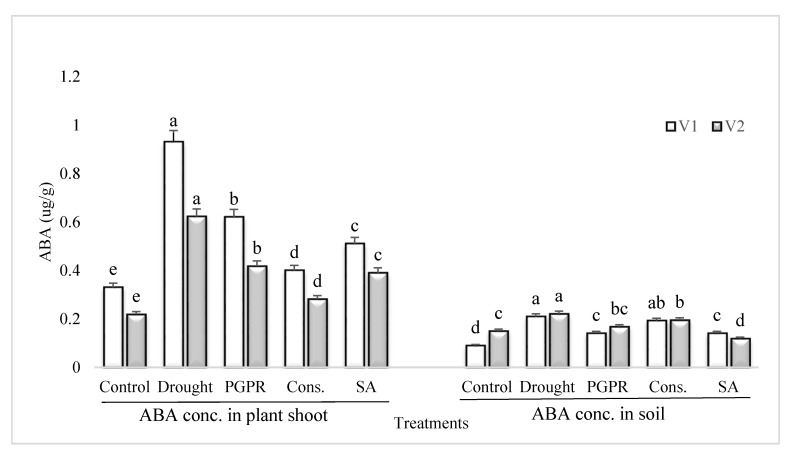
ABA content in plant shoot and rhizosphere of drought sensitive and tolerant varieties grown under rainfed conditions. V1: drought sensitive variety; V2: drought tolerant variety. Different significance levels were denoted with different letters (a, b, c, d and e).

**Figure 6 microorganisms-08-01018-f006:**
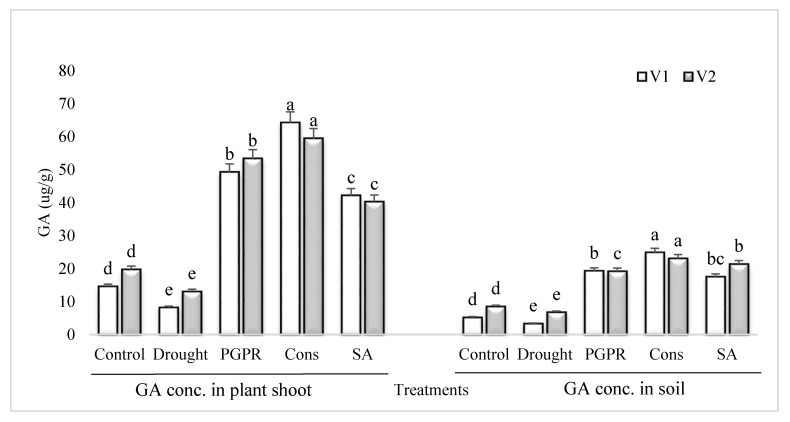
GA content in plant shoot and rhizosphere of drought sensitive and tolerant varieties grown under rainfed conditions. V1: drought sensitive variety; V2: drought tolerant variety. Different significance levels were denoted with different letters (a, b, c, d and e).

**Table 1 microorganisms-08-01018-t001:** Treatments made.

Control (+ve)	Plants grown under normal condition (irrigated)
Control (−ve)	Untreated uninoculated plants grown under rainfed conditions
PGPR	Plants grown under rainfed conditions and treated with *Planomicrobium chinense* and *Bacillus cereus*
Consortium	Plants grown under rainfed conditions and treated with the consortium of 2-PGPR (*Planomicrobium chinense* and *Bacillus cereus*) and Salicylic acid
SA	Plants treated only with Salicylic acid (150 mg/L)

**Table 2 microorganisms-08-01018-t002:** Chlorophyll fluorescence (*Fv*/*Fm* ratio) and performance index (PItot) of two maize varieties under different treatments.

Treatments	Chlorophyll Fluorescence (Fv/Fm)		Performance Index (PI_tot_)	
	V1 ± SE	V2 ± SE	V1 ± SE	V2 ± SE
Control	0.88 ± 0.007	0.95 ± 0.015	23.6 ± 0.15	24.2 ± 0.28
Drought	0.44 ± 0.011	0.65 ± 0.013	11.3 ± 0.09	17.6 ± 0.14
PGPR	0.76 ± 0.006	0.76 ± 0.007	19.2 ± 0.13	21.8 ± 0.21
Consortium	0.91 ± 0.019	0.99 ± 0.012	22.1 ± 0.19	23.4 ± 0.25
SA	0.74 ± 0.008	0.81 ± 0.018	17.6 ± 0.18	20.9 ± 0.17

V1: drought sensitive variety; V2: drought tolerant variety; ± SE-represents standard error values.

**Table 3 microorganisms-08-01018-t003:** Micronutrient accumulation (mg/kg) in the shoot and rhizosphere of sensitive variety grown under rainfed conditions.

Treatments	Copper (Cu)		Cobalt (Co)		Iron (Fe)		Zinc (Zn)	
	Shoot ± SE	Soil ± SE	Shoot ± SE	Soil ± SE	Shoot ± SE	Soil ± SE	Shoot ± SE	Soil ± SE
Control	9.11 ± 0.22	0.94 ± 0.09	5.243 ± 0.21	0.451 ± 0.011	127.1 ± 5.08	0.562 ± 0.003	0.291 ± 0.005	0.086 ± 0.0005
Drought	4.98 ± 0.13	0.33 ± 0.004	1.29 ± 0.06	0.172 ± 0.003	45.7 ± 2.98	0.193 ± 0.001	0.099 ± 0.0001	0.019 ± 0.0001
PGPR	13.43 ± 0.19	2.21 ± 0.08	17.22 ± 0.12	1.13 ± 0.08	268.8 ± 9.11	3.12 ± 0.21	0.314 ± 0.003	1.11 ± 0.003
Consortium	15.71 ± 0.14	4.11 ± 0.11	24.88 ± 0.27	1.42 ± 0.11	302.4 ± 17.05	3.87 ± 0.19	0.454 ± 0.009	1.81 ± 0.001
SA	11.23 ± 0.1	2.01 ± 0.15	14.64 ± 0.23	1.22 ± 0.14	194.3 ± 7.65	2.43 ± 0.13	0.294 ± 0.002	0.097 ± 0.002

± SE-represents standard error values.

**Table 4 microorganisms-08-01018-t004:** Micronutrient accumulation (mg/kg) in the shoot and rhizosphere of tolerant variety grown under rainfed conditions.

Treatments	Copper (Cu)		Cobalt (Co)		Iron (Fe)		Zinc (Zn)	
	Shoot ± SE	Soil ± SE	Shoot ± SE	Soil ± SE	Shoot ± SE	Soil ± SE	Shoot ± SE	Soil ± SE
Control	11.31 ± 0.021	1.27 ± 0.06	8.01 ± 0.41	0.621 ± 0.011	169.4 ± 8.22	0.886 ± 0.06	0.299 ± 0.016	0.083 ± 0.006
Drought	8.65 ± 0.011	0.67 ± 0.002	2.14 ± 0.22	0.328 ± 0.004	98.6 ± 5.12	0.376 ± 0.001	0.176 ± 0.01	0.032 ± 0.002
PGPR	18.45 ± 0.028	2.98 ± 0.012	17.85 ± 1.83	1.1 ± 0.021	322.9 ± 14.91	3.43 ± 0.21	0.342 ± 0.019	1.17 ± 0.008
Consortium	20.1 ± 0.03	4.85 ± 0.017	22.67 ± 2.01	1.27 ± 0.029	376.5 ± 19.01	4.1 ± 0.29	0.385 ± 0.021	1.74 ± 0.011
SA	15.2 ± 0.019	2.52 ± 0.019	11.34 ± 0.91	1.14 ± 0.02	169.7 ± 6.13	2.14+0.14	0.236 ± 0.01	0.088 ± 0.009

± SE-represents standard error values.

**Table 5 microorganisms-08-01018-t005:** Macronutrient accumulation (mg/kg) in the shoot and rhizosphere of sensitive variety grown under rainfed conditions.

Treatments	Calcium (Ca)		Magnesium (Mg)		Potassium (K)		Sodium (Na)	
	Shoot ± SE	Soil ± SE	Shoot ± SE	Soil ± SE	Shoot ± SE	Soil ± SE	Shoot ± SE	Soil ± SE
Control	5.33 ± 0.31	0.057 ± 0.004	1.91 ± 0.055	0.0095 ± 0.0007	9.81 ± 0.47	0.07 ± 0.0003	0.92 ± 0.033	0.0083 ± 0.0003
Drought	2.19 ± 0.17	0.009 ± 0.0005	0.74 ± 0.039	0.0034 ± 0.0004	4.53 ± 0.27	0.009 ± 0.0001	0.38 ± 0.014	0.0014 ± 0.0001
PGPR	9.14 ± 0.47	0.095 ± 0.006	3.94 ± 0.19	0.028 ± 0.003	12.31 ± 0.66	1.23 ± 0.044	1.45 ± 0.064	0.0145 ± 0.013
Consortium	11.74 ± 0.62	1.12 ± 0.042	3.15 ± 0.12	0.019 ± 0.001	12.11 ± 0.59	1.17 ± 0.036	1.93 ± 0.071	0.0201 ± 0.015
SA	6.93 ± 0.46	0.074 ± 0.003	1.98 ± 0.049	0.009 ± 0.0006	9.92 ± 0.38	0.058 ± 0.002	1.13 ± 0.041	0.0101 ± 0.009

± SE-represents standard error values.

**Table 6 microorganisms-08-01018-t006:** Macronutrient accumulation (mg/kg) in the shoot and rhizosphere of tolerant variety grown under rainfed conditions.

Treatments	Calcium (Ca)		Magnesium (Mg)		Potassium (K)		Sodium (Na)	
	Shoot ± SE	Soil ± SE	Shoot ± SE	Soil ± SE	Shoot ± SE	Soil ± SE	Shoot ± SE	Soil ± SE
Control	5.95 ± 0.43	0.081 ± 0.004	1.9 ± 0.14	0.0083 ± 0.0004	10.3 ± 0.67	0.092 ± 0.007	1.12 ± 0.11	0.012 ± 0.001
Drought	3.76 ± 0.29	0.011 ± 0.001	1.18 ± 0.11	0.0063 ± 0.0002	6.85 ± 0.49	0.032 ± 0.003	0.59 ± 0.023	0.0056 ± 0.0003
PGPR	8.19 ± 0.72	0.094 ± 0.006	3.76 ± 0.27	0.023 ± 0.0012	14.22 ± 0.93	1.41 ± 0.11	1.33 ± 0.28	0.0134 ± 0.001
Consortium	12.85 ± 0.83	1.19 ± 0.01	3.51 ± 0.23	0.02 ± 0.001	14.81 ± 1.22	1.5 ± 0.16	1.89 ± 0.41	0.0192 ± 0.002
SA	7.28 ± 0.55	0.079 ± 0.008	1.73 ± 0.13	0.01 ± 0.0001	9.84 ± 0.6	0.053 ± 0.005	1.19 ± 0.12	0.012 ± 0.001

± SE-represents standard error values.

**Table 7 microorganisms-08-01018-t007:** Relative water content (%), soil moisture content (%), electrical conductivity and soil pH.

Treatments	Relative Water Content (RWC)		Soil Moisture Content (SMC)		Electrical Conductivity (µS/cm)		Soil pH	
	V1 ± SE	V2 ± SE	V1 ± SE	V2 ± SE	V1 ± SE	V2 ± SE	V1 ± SE	V2 ± SE
Control	77.7 ± 4.55	84.9 ± 5.78	2.4 ± 0.18	2.7 ± 0.23	100 ± 7.23	110 ± 7.81	7.3 ± 0.38	7.5 ± 0.52
Drought	34.5 ± 2.91	58.3 ± 4.21	0.32 ± 0.019	1.4 ± 0.05	80 ± 5.98	90 ± 6.34	5.6 ± 0.23	5.9 ± 0.25
PGPR	57.7 ± 3.13	61.2 ± 4.61	1.91 ± 0.11	2.31 ± 0.21	130 ± 8.38	110 ± 7.69	8.7 ± 0.61	9.2 ± 0.71
Consortium	72.3 ± 4.22	69.3 ± 5.11	2.3 ± 0.17	2.71 ± 0.33	110 ± 6.91	120 ± 8.09	8.4 ± 0.54	8.1 ± 0.49
SA	49.9 ± 3.01	56.6 ± 3.91	1.6 ± 0.09	1.52 ± 0.13	90 ± 7.11	100 ± 6.97	7.7 ± 0.4	7.9 ± 0.54

V1: drought sensitive variety; V2: drought tolerant variety, ± SE-represents standard error values.

**Table 8 microorganisms-08-01018-t008:** Soil organic carbon (SOC), total nitrogen (TN) and bulk density of rhizosphere soil.

Treatments	SOC (%)		Total Nitrogen (%)		C/N Ratio		Soil Bulk Density (g/cm^3^)	
	V1 ± SE	V2 ± SE	V1 ± SE	V2 ± SE	V1 ± SE	V2 ± SE	V1 ± SE	V2 ± SE
Control	3.12 ± 0.17	3.19 ± 0.21	0.25 ± 0.013	0.26 ± 0.021	14.22 ± 0.081	14.91 ± 1.1	1.33 ± 0.13	1.31 ± 0.11
Drought	1.94 ± 0.08	2.15 ± 0.13	0.16 ± 0.008	0.19 ± 0.012	9.13 ± 0.067	11.7 ± 0.043	1.81 ± 0.11	1.75 ± 0.17
PGPR	3.32 ± 0.23	3.51 ± 0.23	0.25 ± 0.011	0.28 ± 0.019	15.1 ± 0.089	15.31 ± 0.094	1.16 ± 0.09	1.13 ± 0.07
Consortium	3.39 ± 0.29	3.7 ± 0.28	0.28 ± 0.017	0.3 ± 0.031	15.81 ± 1.11	15.93 ± 1.21	1.01 ± 0.04	0.97 ± 0.03
SA	3.21 ± 0.19	3.39 ± 0.21	0.23 ± 0.014	0.21 ± 0.011	12.23 ± 0.054	12.41 ± 0.066	1.21 ± 0.014	1.17 ± 0.09

V1: drought sensitive variety; V2: drought tolerant variety; SOC: soil organic content; C/N: carbon–nitrogen ratio; ± SE-represents standard error values.

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
