# Peer review of "Role of Beneficial Microorganisms and Salicylic Acid in Improving Rainfed Agriculture and Future Food Safety"

_microorganisms, 2020, doi:10.3390/microorganisms8071018_

Round 1

Reviewer 1 Report

The Manuscript ID microorganisms-830367 by Khan and Bano entitled “Role of beneficial microorganisms and salicylic acid in improving rainfed agriculture and future food safety” reports on results from an investigation testing the hypothesis that inoculation of seeds with a PGPR consortium and a foliar treatment with salicylic acid could induce in maize a better tolerance to water scarcity. The issue addressed by the research is surely interesting, but the MS needs deep revision before a possible evaluation of the reliability of the reported results and of their consistency in sustaining the conclusions drawn by the authors.

The main critical points that make the MS unsuitable even for a first evaluation are listed in the following lines.

In the Material and Methods section, no detail is reported about most of the measurements and/or analyses carried out. In detail, no information is reported about: a) the PRGP consortium used; b) the PRPG seed inoculation procedure adopted; c) the salicylic acid treatment conducted (i.e., at what plant developmental stage, what concentration of salicylic acid, how was SA applied, etcetera); d) at what plant developmental stage the physiological and biochemical analyses were carried out and how many days after salicylic acid treatment); e) total shoot biomass produced in the different conditions; f) by what methods the different phytohormones (IAA, GA, ABA), as well as micro- and macronutrients, were extracted from plants or soil and analyzed; g) how the soil rhizosphere was collected. Without this basic information, it is not possible to evaluate the reliability of the reported results.

Moreover, some of the results seem to be not immediately related to the overall mainstream of the MS. Probably, reconsidering the Introduction section by limiting unnecessary details (i.e. those concerning maize quality as food and feed) and including information on the existing knowledge about the effects of PGRP on the rhizosphere environment and on plant physiological processes, many of the reported results would gain meaning.

Figures are largely unclear. In most of them, neither the titles of the axes nor the units of measure are correctly reported. Just as examples, how is it possible that chlorophyll concentration and performance index have the same scale and unit of measure (Fig. 1)?, Likewise, in Fig.2 it is not possible to use the same y-axis for the Fv/Fm ratio (a number) and the carotenoids concentration (µg/g).

For what reason it has been evaluated the concentration of soluble proteins if the activities of antioxidant enzymes (Fig. 3) are expressed on a tissue fresh weight basis and not as specific activities, which would be advisable if one considers the relative water content of shoots as different as a function of the different treatments? A similar question rises for sugar analysis, for which I could not find in the MS any related result.

Several other critical points are present in the MS, but currently, I do believe that before being able constructively to review the paper it needs to be deeply revised and rewritten considering the above points.

Author Response

The Manuscript ID microorganisms-830367 by Khan and Bano entitled “Role of beneficial microorganisms and salicylic acid in improving rainfed agriculture and future food safety” reports on results from an investigation testing the hypothesis that inoculation of seeds with a PGPR consortium and a foliar treatment with salicylic acid could induce in maize a better tolerance to water scarcity. The issue addressed by the research is surely interesting, but the MS needs deep revision before a possible evaluation of the reliability of the reported results and of their consistency in sustaining the conclusions drawn by the authors.

Response: We would like to thank the Reviewer for his/her evaluation and for the constructive comments and suggestions that have helped us improve the quality of the manuscript. We have revised the manuscript following your suggestions and comments to improve its quality. All the changes made in responses to the Reviewer’s comments were tracked in the revised file of the manuscript. We hope that our revised version will now meet your expectations. Please see below our responses itemized to your comments and suggestions.

The main critical points that make the MS unsuitable even for a first evaluation are listed in the following lines.

In the Material and Methods section, no detail is reported about most of the measurements and/or analyses carried out. In detail, no information is reported about: a) the PRGP consortium used; b) the PRPG seed inoculation procedure adopted; c) the salicylic acid treatment conducted (i.e., at what plant developmental stage, what concentration of salicylic acid, how was SA applied, etcetera); d) at what plant developmental stage the physiological and biochemical analyses were carried out and how many days after salicylic acid treatment); e) total shoot biomass produced in the different conditions; f) by what methods the different phytohormones (IAA, GA, ABA), as well as micro- and macronutrients, were extracted from plants or soil and analyzed; g) how the soil rhizosphere was collected. Without this basic information, it is not possible to evaluate the reliability of the reported results.

Response: We highly appreciate the reviewer for these constructive comments, however, most of the above mentioned basic information are already present in the MS.

  1. a) The consortium was a combined treatment of 2-PGPRs (i.e. Planomicrobium chinense and Bacillus cereus) + Salicylic acid. These information were already present under the subsection Experimental work plan and now these information are put in Table 1.
  2. b) The seed inoculation procedure is already mentioned in the MS under subheading 2.2. as ‘’ The seeds were successively washed in autoclaved distilled water followed by soaking in broth culture of PGPR for 3 hours prior to sowing. There were 1 × 105 colonies of bacteria per seed’’.
  3. c) These all information are already present in the MS as ‘’The foliar spray of SA (150 mg/L), was applied on 28-days old seedlings’’ (i.e. at three leaf stage) at lines 17 and 18.
  4. d) The physiological and biochemical analysis was carried out after one week of Salicylic acid application (i.e. 35 days after sowing). This information is added in the MS at line 124.
  5. f) We adopted the following procedures for the determination of phytohormones in shoot and soil and for micro and macronutrients analysis. We have also added these missing information in the M&M:

2.3.5. Determination of IAA, GA and ABA Contents of Maize Leaves and Rhizosphere Soil

Plant hormones were extracted following the method of Kettner and Dörffling [35]. Fresh plant leaves (1 g) were grinded in 80% methanol at 4 0C with butylated hydroxytoluene @ 10 mg/L used as an antioxidant. The samples were analysed on High Performance Liquid Chromatography (HPLC). For identification of hormones, 100 µL samples filtered through Millipore (SPEC) filter were injected into the column. Commercially grade Abscisic acid (ABA), Indole acetic acid (IAA) and Gibberellic acid (GA), were used as reference for the retention time and peak area measured at 280 nm and 254 nm respectively. For ABA the samples were injected on to a C18 column and eluted at 254 nm with a linear gradient of methanol (30-70%), containing 0.01% acetic acid [36]. Whereas, the phytohormones contents of soil were analyzed using the method of Hartung et al. [37]. For this purpose, the soil samples were extracted in 3 fold excess of 1Mm CaCl2 for an hour. The extraction is then partitioned thrice with ethyl acetate followed by drying of ethyl acetate with the help of rotary thin film evaporator (RFE). The dried sample was dissolved in 1mL methanol (100%) and analyzed for the presence of phytohormones using HPLC as described above.

2.5. Nutrient Analyses of Plant Extracts

Oven dried leaves sample (0.25 g) were taken in 50 mL flask to which 6.5 mL of mixed acid solution, i.e., Nitric acid, Sulphuric acid, Perchloric acid (5:1:0.1) were added and boiled on hot plate under fume hood till the digestion has been completed which was indicated by white fumes coming out from the flasks. Thereafter, few drops of distilled water were added and allowed to cool. The digested samples were transferred in 50 mL volumetric flasks and the volume was made up to 50 mL by adding distilled water. The extract was filtered with Whatmann No. 42 filter paper and concentration of these elements was determined by Atomic Absorption Spectrophotometer (Shimadzu AA-670).

Stock solution (100 ppm) of different elements were prepared for the determination of macro and micronutrients following the method of Allen et al. [41].

  1. g) Soil samples were collected from the rhizosphere of maize at a depth of 10 cm and packed in a Ziploc bag. The collected samples were sent to laboratory within the few hours and stored at -80 ºC.

Moreover, some of the results seem to be not immediately related to the overall mainstream of the MS. Probably, reconsidering the Introduction section by limiting unnecessary details (i.e. those concerning maize quality as food and feed) and including information on the existing knowledge about the effects of PGRP on the rhizosphere environment and on plant physiological processes, many of the reported results would gain meaning.

Response: We would like to thank the reviewer for these constructive comments. Based on these comments we have added the following paragraphs to the Introduction section:

Bacteria that colonize plant roots and enhance plant growth are stated as plant growth-promoting rhizobacteria (PGPR). PGPR are extremely diverse group of micro-organism that occur in the vicinity of plant roots [7]. They significantly influence plant health as they enhance plant tolerance to various stresses and improve the fertility status of rhizosphere soil.  Rhizosphere soil acts as a fount of all nutrients needed by plants for growth. The three most common nutrients are nitrogen, phosphorus, and potassium followed by calcium, magnesium and sulfur. Plants also need small quantities of iron, manganese, zinc, copper, boron and molybdenum [8]. However, stressful environments result in unusually low or high soil pH which in turn results in unavailability of certain nutrients to the plants. PGPR improve the availability and mobilization of macro and micronutrients in rhizosphere and make them available to the host plant. They are well-known to mend the effectiveness of plants and responses to various stimuli [9]. PGPR induced proliferation of root system and improvement in root architecture have been the primary effects [10, 11].

Previous studies have reported that PGPR affects the physiology of plants to attenuate the adverse effects of both the biotic and abiotic stresses [12, 13]. These microorganisms regulate the water potential and stomatal opening through modulating the hydraulic conductivity and transpiration rate. Marulanda et al. [14] reported that maize plants inoculated with Bacillus sp., showed increase in root hydraulic conductivity as compared to uninoculated plants grown under stress conditions. Colonization by PGPR is associated with changes in plant metabolism, signaling and hormone homeostasis. Different PGPR strains can synthesize phytohormones, metabolize them, or affect plants' hormone synthesis and signal transduction [15]. PGPR that produce auxins have been shown to elicit transcriptional changes in hormone, defense-related, and cell wall related genes [16], induce longer roots [17], increase root biomass and decrease stomata size and density [18], and activate auxin response genes that enhance plant growth [19]. Some strains of PGPR can promote relatively large amounts of gibberellins, leading to enhanced plant shoot growth [20]. Interactions of these hormones with auxins can alter root architecture [21]. PGPR induce changes in phytohormone signaling and osmolyte accumulation facilitate plants to grow well under stressful environments. Inoculation with B. subtilis increase photosynthesis in Arabidopsis through the modulation of plant endogenous sugar and abscisic acid (ABA) signaling, with a regulatory role for plant symbionts in photosynthesis [22]. Shi et al. [23] showed that endophytic bacteria species increased the photosynthetic capacity and total chlorophyll content of sugar beet, leading to a consequent increased carbohydrate synthesis, these increases were promoted by phytohormones, which were produced by the bacteria. Beside this, beneficial microorganisms can stimulate carbohydrate metabolism and transport, which directly implicate source-sink relations, photosynthesis, growth rate and biomass reallocation. Seed inoculated with B. aquimaris strains increased total soluble sugars and reduced sugars in wheat under saline field conditions and resulted in higher shoot biomass and NPK accumulation [24]. PGPR also play an important role in enhancing the fertility status of rhizosphere through various mechanisms, such as decomposition of crop residues, mineralization of soil organic matter, immobilization of mineral nutrients, phosphate solubilizers, synthesis of soil organic matter, nitrification, nitrogen fixation, and plant growth promoters including nutrient acquisition phytohormone production (biostimulants), and suppression of plant pathogens [25]. In contrast to PGPR, salicylic acid are chemical messengers significantly affecting the plant responses and water budget under stressful environments

Figures are largely unclear. In most of them, neither the titles of the axes nor the units of measure are correctly reported. Just as examples, how is it possible that chlorophyll concentration and performance index have the same scale and unit of measure (Fig. 1)?, Likewise, in Fig.2 it is not possible to use the same y-axis for the Fv/Fm ratio (a number) and the carotenoids concentration (µg/g).

REAPONSE: We highly appreciate the reviewer for these important comments. Based on these comments, we have converted the results of chlorophyll fluorescence and performance index in tabulated form (Table 2).

For what reason it has been evaluated the concentration of soluble proteins if the activities of antioxidant enzymes (Fig. 3) are expressed on a tissue fresh weight basis and not as specific activities, which would be advisable if one considers the relative water content of shoots as different as a function of the different treatments? A similar question rises for sugar analysis, for which I could not find in the MS any related result.

Response: We are thankful to the reviewer for pointing out this mistake. We have removed these extra details from M&M.

Several other critical points are present in the MS, but currently, I do believe that before being able constructively to review the paper it needs to be deeply revised and rewritten considering the above points.

Response: Based on these comments, we have made significant changes to our MS and we are hoping that our revised version would now meet your expectations.

Reviewer 2 Report

The manuscript is interesting, stimulating and very current. For this reason, I believe that the manuscript is suitable for publication in the journal. The abstract is written in general. Instead of percentages, it would be more appropriate to use measured values and percentages in addition to decreases or increases. I believe that the introduction does not fully correspond to the focus of the manuscript. I recommend to complement the characteristics of the soil as nutrients were evaluated in the experiment. It is not entirely clear design experiment. Perhaps it would be better to put it on the table. I recommend adding variety designations to the legends of all graphs. Why are carotenoid content values included in the fluorescence graph? The tables would be appropriate to complement the standard deviation and statistically relevance. The results are evaluated as a percentage, but the measured or calculated values are in the graphs. It is not possible to clearly determine whether the percentages correspond. I recommend editing. The discussion is rather descriptive. I also recommend unifying and correcting the cited sources.

Author Response

We would like to thank the Reviewer for his/her evaluation and for the constructive comments and suggestions that have helped us improve the quality of the manuscript. We have revised the manuscript following your suggestions and comments to improve its quality. All the changes made in responses to the Reviewer’s comments are tracked in the revised file of the manuscript. We hope that our revised version will now meet your expectations. Please see below our responses itemized to your comments and suggestions.

The manuscript is interesting, stimulating and very current. For this reason, I believe that the manuscript is suitable for publication in the journal. The abstract is written in general. Instead of percentages, it would be more appropriate to use measured values and percentages in addition to decreases or increases.

RESPONSE: We highly appreciate the reviewer for this constructive comment. Based on this comment, the percent increases/decreases of various parameters are added in the abstract.

I believe that the introduction does not fully correspond to the focus of the manuscript. I recommend to complement the characteristics of the soil as nutrients were evaluated in the experiment.

Response: We are thankful to the reviewer for this constructive comment. We have extensively modified the introduction and added the following paragraphs to Introduction:

Bacteria that colonize plant roots and enhance plant growth are stated as plant growth-promoting rhizobacteria (PGPR). PGPR are extremely diverse group of micro-organism that occur in the vicinity of plant roots [7]. They significantly influence plant health as they enhance plant tolerance to various stresses and improve the fertility status of rhizosphere soil.  Rhizosphere soil acts as a fount of all nutrients needed by plants for growth. The three most common nutrients are nitrogen, phosphorus, and potassium followed by calcium, magnesium and sulfur. Plants also need small quantities of iron, manganese, zinc, copper, boron and molybdenum [8]. However, stressful environments result in unusually low or high soil pH which in turn results in unavailability of certain nutrients to the plants. PGPR improve the availability and mobilization of macro and micronutrients in rhizosphere and make them available to the host plant. They are well-known to mend the effectiveness of plants and responses to various stimuli [9]. PGPR induced proliferation of root system and improvement in root architecture have been the primary effects [10, 11].

Previous studies have reported that PGPR affects the physiology of plants to attenuate the adverse effects of both the biotic and abiotic stresses [12, 13]. These microorganisms regulate the water potential and stomatal opening through modulating the hydraulic conductivity and transpiration rate. Marulanda et al. [14] reported that maize plants inoculated with Bacillus sp., showed increase in root hydraulic conductivity as compared to uninoculated plants grown under stress conditions. Colonization by PGPR is associated with changes in plant metabolism, signaling and hormone homeostasis. Different PGPR strains can synthesize phytohormones, metabolize them, or affect plants' hormone synthesis and signal transduction [15]. PGPR that produce auxins have been shown to elicit transcriptional changes in hormone, defense-related, and cell wall related genes [16], induce longer roots [17], increase root biomass and decrease stomata size and density [18], and activate auxin response genes that enhance plant growth [19]. Some strains of PGPR can promote relatively large amounts of gibberellins, leading to enhanced plant shoot growth [20]. Interactions of these hormones with auxins can alter root architecture [21]. PGPR induce changes in phytohormone signaling and osmolyte accumulation facilitate plants to grow well under stressful environments. Inoculation with B. subtilis increase photosynthesis in Arabidopsis through the modulation of plant endogenous sugar and abscisic acid (ABA) signaling, with a regulatory role for plant symbionts in photosynthesis [22]. Shi et al. [23] showed that endophytic bacteria species increased the photosynthetic capacity and total chlorophyll content of sugar beet, leading to a consequent increased carbohydrate synthesis, these increases were promoted by phytohormones, which were produced by the bacteria. Beside this, beneficial microorganisms can stimulate carbohydrate metabolism and transport, which directly implicate source-sink relations, photosynthesis, growth rate and biomass reallocation. Seed inoculated with B. aquimaris strains increased total soluble sugars and reduced sugars in wheat under saline field conditions and resulted in higher shoot biomass and NPK accumulation [24]. PGPR also play an important role in enhancing the fertility status of rhizosphere through various mechanisms, such as decomposition of crop residues, mineralization of soil organic matter, immobilization of mineral nutrients, phosphate solubilizers, synthesis of soil organic matter, nitrification, nitrogen fixation, and plant growth promoters including nutrient acquisition phytohormone production (biostimulants), and suppression of plant pathogens [25]. In contrast to PGPR, salicylic acid are chemical messengers significantly affecting the plant responses and water budget under stressful environments

 It is not entirely clear design experiment. Perhaps it would be better to put it on the table.

Response: We highly appreciate the Reviewer for this important suggestion. The experimental work plan is converted to tabulated form (Table 1).

 I recommend adding variety designations to the legends of all graphs. 

Response: We highly appreciate the Reviewer for this important comment. We have added these details to the legends of all figures.

Why are carotenoid content values included in the fluorescence graph? The tables would be appropriate to complement the standard deviation and statistically relevance. The results are evaluated as a percentage, but the measured or calculated values are in the graphs. It is not possible to clearly determine whether the percentages correspond. I recommend editing.

Response: We highly appreciate the Reviewer for this important recommendation. Based on these comments we separated the graph of carotenoid from Chl fluorescence and converted the graphs for Chl fluorescens and Performance index to tabulated form (Table 2).

 The discussion is rather descriptive. 

Response: We are thankful to the reviewer for this constructive comment. Based on these comments we significantly modified the discussion section as follows:

Rainfed agriculture comprises more than 75% of the cultivated area of the world. In Pakistan more than 25% of the total area is designated as rainfed and greater than 30% population is dependent on rainfed agriculture. Furthermore, these regions are badly effected by frequent droughts that adversely effects the plant physiology and productivity consequently affecting the livelihood of the inhabitant. Low productivity in these regions is accompanied by limited water supply, degraded soil health associated with low fertility and limited supply of nutrients. The present investigation yielded valuable information pertaining to the phytohormonal status of soil which originate from the root exudates of the growing plants also synthesized from soil microbiota. This information may be useful for successive planting.

                Rainfed conditions adversely affected the growth of plants and photosynthetic efficiency as evidenced by the significant decrease in the chlorophyll and carotenoids contents, chlorophyll fluorescence and performance index (PI). The sensitive variety was affected more. Previous studies document significant reductions in these parameters in plants grown under rainfed conditions [42]. The ameliorative effects of PGPR was more pronounced than SA for the increase in the contents of chlorophyll and carotenoids. The observed stimulatory effects was further augmented in the combined treatment may be attributed to the fact that PGPR and SA significantly improved the relative water content of the plants grown under rainfed condition, demonstrating the positive relationship between RWC and chlorophyll fluorescence. The effect being more pronounced in the sensitive variety and the plants were more responsive to the combined treatment of consortium of PGPR and SA. Noteworthy, the drought induced decrease in carotenoids content was much greater in the sensitive variety than tolerant variety. Carotenoids act as protective pigment and scavengers of ROS [43] thereby enhancing the chlorophyll fluorescence and performance index. As potential antioxidants, they are essential in different plant processes under stress. They act as light harvesters, and scavenge the triplate state chlorophylls and singlet oxygen species, depletes excess harmful energy during stress conditions and membrane stabilizers [44]. Carotenoids also play an important role in the mechanisms protecting the photosynthetic apparatus against various harmful environmental factors [45] thus ensuring enhanced plant growth under unusual circumstances. SA was considered to prevent the degradation of photochemical pigments under stress condition [46, 47]. The higher chlorophyll fluorescence is an indicator for detection of plant tolerance to various stresses. PI is a sensitive indicator of water scarcity in plants [48, 49]. The applications of SA and PGPR, assisted plants in water and nutrient uptake and thus maintained higher chlorophyll fluorescence and PI under rainfed conditions. The combined treatment of PGPR and SA significantly improved the water budget of plants, which eventually ensued higher photochemical efficiency and growth of plants of both the varieties compared to untreated plants [50-53]. 

Increased level of antioxidant enzymes such as CAT, POD and APOX was exhibited in the current study demonstrating better potential for scavenging the reactive oxygen species (ROS) also reported previously in wheat [54], barley [55], rice [56], soybean [57] and chickpea [58] under drought condition. The PGPR and SA significantly reduced the activities of antioxidant enzymes as compared to untreated plants grown under rainfed conditions. This decrease could be attributed to the fact that application of PGPR in combination with SA overcame the oxidative stress which is produced as secondary stress under drought stress thereby the production of ROS is minimized hence, the production of extra antioxidant enzyme activity ;may not be required. Khan et al. [59] demonstrated PGPR induced reduction in the activities of antioxidant enzymes in chickpea shoot grown in sandy soil condition. Reduction in antioxidant enzymes activity by the application of SA had also been reported previously [60]. PGPR and SA decreased the lipid peroxidation as measured by the malondialdehyde content. The content of lipid peroxidation increases in plants with the increase in ROS, thereby, adversely affecting the physiological and biochemical process in plants under stress. Higher content of lipid peroxidation also adversely affects permeability of cell membranes, ion transport, enzymatic activity and protein cross-talk thus disrupting overall cellular metabolism and eventually lead to cell death [61]. PGPR reduced oxidative damages by reducing lipid peroxidation had been reported earlier by Habibzadeh et al. [62] in canola and by Sahin et al. [63] in lettuce plants grown under waterlogged conditions.

Plant growth and development is controlled by phytohormones and the PGPR-based direct promontory effects involved in the biosynthesis of IAA and GA in plants [64]. Drought induced decrease in IAA was higher in the sensitive variety as compared to control. The tolerant variety resisted the adverse effect of drought to a greater extent. All the treatments have ameliorative effects but SA treatment showed lower potency in this context, could be due to the basic difference between the PGPR and SA in the biosynthesis or modulation of phytohormones. PGPR–induced increase in IAA was greater in the sensitive variety. Similar pattern of response was recorded in the GA content of rhizosphere soil. The rhizosphere soil demonstrated the residual effects of growing drought stressed plants and PGPR treated plants. The drought induced decrease in IAA content was alleviated by PGPR. PGPR consortium assisted SA in combination.GA is an important plant growth promoting hormone responsible for cell elongation and subsequently higher biomass production [65]. GA content was several folds higher in plants treated with PGPR alone or more so in consortium. This is an adoptive mechanism to combat the adverse effect of drought (moisture stress) and promote the growth and biomass production in plants. The SA was less effective than the PGPR/PGR consortium. Phytohormones produced by root-associated microbes may be important target for metabolic engineering host plant to induce tolerance to abiotic stresses. 

As documented previously in all stresses the endogenous ABA level increases significantly to impart tolerance to stresses [66]. Noteworthy the drought induced increase in ABA content was higher in the sensitive variety. The PGPR and SA effects to decrease drought induced augmentation in ABA was higher in the tolerant variety this may be attributed to the fact that drought induced decrease in soil moisture and RWC was lesser in the tolerant variety. The significant increase in ABA content in plants under rainfed condition particularly in the sensitive variety was counteracted by PGPR and SA application. PGPR used are EPS producing bacteria and also function in water conservation hence, reduced significant increase of ABA as higher ABA exert inhibitory effects on plant productivity and stomatal conductance is reduced [67, 68]. However some increase in ABA is required to combat stress as ABA also acts as stress hormone and ameliorate the osmotic and oxidative stresses in plants which are created under moisture deficit conditions in soil. ABA is considered to accumulate in the roots and transported along with the xylem sap to the leaves [69]. However, the PGPR particularly the consortium of PGPR in combination with SA enhance the ABA content greater than control this aimed to improve the water holding capacity of soil and to minimize the evapotranspiration and the water loss [66]. The effect of PGPR inoculation and that of SA was at par but the consortium was even more stimulatory and the % increase was higher in the tolerant variety correlated with the soil moisture retained in the tolerant variety. The phytohormone analysis of rhizosphere soil demonstrated the significant increase in ABA content possibly originating from the root exudate of stressed plants in addition being synthesized by the associated microbes. This warrant the inhibition in germination and seedling establishment of the succeeding crop. Nevertheless, the rhizosphere of PGPR or SA treated plants exhibited decrease in drought induced ABA accumulation. Thus the soil can have better potential for seed germination and seedling establishment as consortium increased IAA, GA content while decreased the ABA content.

Relative water content of leaves (RWC) and soil moisture content (SMC) are indicators of water status of plant and rhizosphere soil respectively under drought stress condition [70]. The application of PGPR alone and more so in combination with SA alleviated the decrease in RWC and SMC and significantly improved the RWC and SMC of both the varieties grown under rainfed conditions. Higher RWC enable plants to recover from stresses and improve its growth and yield stability [71, 72]. Kasim et al. [73] and Mahmood et al. [74] reported significant increases in the RWC of barley plants inoculated with biofilm forming plant growth promoting rhizobacteria. Availability of soil moisture is prerequisite for seed germination, root development and nutrient uptake [75]. Naveed et al. [76] reported that bacterial inoculation increases the relative water content in plants and decreases the electrolyte leakage leading to drought tolerance in plants. The moisture deficit condition of rainfed soil decreases soil moisture which was alleviated and taken as par to the control by the application of consortium of PGPR and SA; more pronounced in the tolerant variety (V2).

PGPR release various chelating substances in the rhizosphere that affect the availability and mobility of essential nutrients [77]. The inoculation of plants with. P. chinense and B. cereus significantly improved the accumulation of macro- and micronutrients in plant shoot and rhizosphere soil of both the varieties. The PGPR was more effective and assisted SA when in combination. was more effective and the pronounced increase was recorded in the contents of Ca, K, Mg, Na, Cu, Co, Fe and Zn in the rhizosphere soil and subsequently in shoot of plants under rainfed conditions; the tolerant variety being more responsive. It is further evidenced by the result that the basic level of all the macro and micronutrients were higher in the tolerant variety over that of sensitive variety though the sensitive variety was more responsive to treatments. Iron and Zinc are integral part of some enzymes and pigments and play an important role in the synthesis of DNA, photosynthesis, respiration and assist in the production of energy in plants under extreme climatic conditions [78]. Higher content of Cu in the rhizosphere involved in lignification; provide strength and prevent lodging in cereal crops, the deficiency of Cu in plants leads to poor growth, delayed flowering and sterility and also enhances the susceptibility to diseases [79]. The combined treatment of 2-PGPR augmented Ca, K and Mg accumulation in the shoot as well as in rhizosphere of sensitive variety. Rana et al. [80] reported a significant increase in the nutrient content of plants inoculated with the consortia of 3-PGPR. Aini et al. [81] reported that the inoculation of lettuce plants with PGPR and AMP resulted improved root colonization and macronutrient uptake.

Soil organic matter is the index of soil quality and soil stability [82]. Rainfed conditions adversely effected the soil organic content and nitrogen in the rhizosphere of untreated plants. PGPR and SA significantly improved the soil organic content and total nitrogen in the rhizosphere of both the varieties but the tolerant variety was more responsive. Decomposition of organic matter results in the release of essential nutrients and thus helps in maintaining nutrient cycling. Higher concentration of soil organic content assist in binding soil particles into aggregates, which facilitates water filtration and aeration. Soil organic matter are the major pools of carbon in the biosphere and can act both as a source and sink of carbon [83, 84]. Previously, Hassan and Bano [85] reported that the inoculation of wheat plants with PGPR significantly improved the soil organic content and total nitrogen in rhizosphere soil. Present results are in agreement with those reported by Hassan and Bano [85] in wheat, Xiaohui et al. [86] in tomato plants and Naseri et al. [87] in rapeseed, where the application of PGPR significantly enhanced the soil nitrogen and organic matter. This significant increase in the consortium of PGPR and SA was due to their ability of marinating higher C/N ratio in soil as, C/N ratio affects overall turnover rates of soil organic matter and improve soil structure and plant growth.

I also recommend unifying and correcting the cited sources.

Response: We are thankful to the reviewer for pointing out this mistake. We have revised and unified  all the references.  

Round 2

Reviewer 1 Report

The present version of the MS is way better than the previous one. Excluding a few minor aspects, the authors essentially answered the questions I posed and integrated the MS according to my suggestions.

Nevertheless, some problems remain, but since the authors can easily solve them, I believe that MS, after these minor changes, will be soon suitable for the publication in Microorganisms.

The authors answered some of my questions only in the rebuttal letter. They did not introduce them in the appropriate section(s) of the MS. Please consider this general aspect. In particular: a) details about the SA treatment should be described in the M&M section and not only in the abstract; b) information about the collection of rhizospheric soil, present in the rebuttal letter, should be added also to the M&M section; c) methods about the analyses of carotenoids are still lacking, like those on plant shoot biomass.

Please introduce in all the tables (including the new Tab. 2) the SE (or SD) values for each result.

Fig. 1 is now clear and correct. Please, use it as a model, i.e. introduce the double y-axis (on the left and right sides) also in Fig. 3 and report the units of measure of the dependent variable on the y-axis and not below the x-axis. The latter suggestion concerning units of measure on the y-axis is valid for all the other figures.

Finally, I suggest that when the definitive version is ready, it should anyway be revised for improving a few language and editing details.

Author Response

The present version of the MS is way better than the previous one. Excluding a few minor aspects, the authors essentially answered the questions I posed and integrated the MS according to my suggestions.

Nevertheless, some problems remain, but since the authors can easily solve them, I believe that MS, after these minor changes, will be soon suitable for the publication in Microorganisms.

RESPONSE: Authors are very appreciative to the worthy reviewer. The whole manuscript has been thoroughly revised and improved according to valuable suggestions.

The authors answered some of my questions only in the rebuttal letter. They did not introduce them in the appropriate section(s) of the MS. Please consider this general aspect. In particular: a) details about the SA treatment should be described in the M&M section and not only in the abstract; b) information about the collection of rhizospheric soil, present in the rebuttal letter, should be added also to the M&M section; c) methods about the analyses of carotenoids are still lacking, like those on plant shoot biomass.

RESPONSE:  a): We have incorporated these missing information in the M&M as:

The inoculated maize seeds were sprayed with aqueous solution of SA (150 mg/L) at three leaf stage. Salicylic Acid is a beta hydroxy acid that occurs as a natural compound in plants and is hormonal in action (Lines 117-119).

b) These missing information are now added in the M&M at lines 179-181.

For soil analysis, soil samples were collected from the rhizosphere of maize at a depth of 10 cm and packed in a Ziploc bag. The collected samples were sent to the laboratory within a few hours and stored at -80 ºC.

c) We are thankful to the reviewer for pointing out this mistake, the missing information are added at lines 133-135.

Please introduce in all the tables (including the new Tab. 2) the SE (or SD) values for each result.

RESPONSE: We have added the standard error values in all tables for each result.

Fig. 1 is now clear and correct. Please, use it as a model, i.e. introduce the double y-axis (on the left and right sides) also in Fig. 3 and report the units of measure of the dependent variable on the y-axis and not below the x-axis. The latter suggestion concerning units of measure on the y-axis is valid for all the other figures.

RESPONSE: We are thankful to the reviewer for this important suggestion. We have modified the Fig. 3 based on this suggestion.

Finally, I suggest that when the definitive version is ready, it should anyway be revised for improving a few language and editing details.

RESPONSE: We have revised our MS based on this comment.

Reviewer 2 Report

The manuscript was modified by the authors and its quality improved. A number of comments and remarks were accepted. I have comments on the results again. it would be appropriate to add standard deviations to the tables, as shown in the graphs. The results are given in relative values (%) and not in real numbers. The results should be based on the values given in the graphs and tables. The reader has to recalculate it. It's a matter of opinion.

Author Response

The manuscript was modified by the authors and its quality improved. A number of comments and remarks were accepted. I have comments on the results again. it would be appropriate to add standard deviations to the tables, as shown in the graphs. The results are given in relative values (%) and not in real numbers. The results should be based on the values given in the graphs and tables. The reader has to recalculate it. It's a matter of opinion.

RESPONSE: We are very grateful to the worthy reviewer for this important comment. Based on this comment we have added the standard error values in all tables for each result.